# Genetic Diversity and Population Structure Analysis of Excellent Sugar Beet (*Beta vulgaris* L.) Germplasm Resources

Fei Peng [1,2], Zhi Pi [1,2], Shengnan Li [1,2,*] and Zedong Wu [1,2,*]

1 Academy of Modern Agriculture and Ecological Environment, Heilongjiang University, Harbin 150080, China; pf981124@163.com (F.P.); 2018060@hlju.edu.cn (Z.P.)
2 Key Laboratory of Sugar Beet Genetic Breeding, Heilongjiang University, Harbin 150080, China
* Correspondence: 2020046@hlju.edu.cn (S.L.); 1997009@hlju.edu.cn (Z.W.)

**Abstract:** This study analyzed the genetic diversity, population structure, and cluster analysis of 129 sugar beet germplasm resources to screen superior germplasms for breeding using the 27 simple sequence repeat (SSR) and 33 pairs of insertion–deletion (InDel) molecular markers. After integrating the phenotypic variation of 16 descriptive and 4 qualitative phenotypic variables, the genetic variation levels of the 129 sugar beet germplasms' phenotypic traits were analyzed using the principal component analysis (PCA), correlation analysis, and analysis of variance methods. The genetic diversity examination of molecular markers showed a polymorphism information content (PIC) of 0.419–0.773 (mean = 0.610). Moreover, the mean number of effective alleles detected via the SSR and InDel markers was 3.054 and 2.298, respectively. Meanwhile, the PIC ranged from 0.130 to 0.602 (mean = 0.462). The population structure analysis revealed the most appropriate K-value, indicating three populations (K = 3). The genetic distances of the 129 germplasm resources ranged from 0.099 to 0.466 (mean = 0.283). The cluster analysis results demonstrated that the germplasms were grouped into three primary classes. Based on the analysis of variance, the two qualitative features with the highest coefficients of variation were petiole width (16.64%) and length (17.11%). The descriptive trait root length index (1.395) exhibited the greatest genetic diversity. The PCA reduced the 20 phenotypic traits into five principal components, contributing 51.151%. The results of this study provide a theoretical foundation for the future selection and breeding of superior sugar beet germplasm resources.

**Keywords:** sugar beet; genetic diversity; population structure; phenotypic traits; molecular markers; breeding

## 1. Introduction

The sugar beet (*Beta vulgaris* L.) is a biennial plant belonging to the Amaranthaceae family. The sea beet (*Beta vulgaris* ssp. *maritima*) is a wild ancestor of the sugar beet that originates from the Mediterranean region [1]. Sugar beet is one of the largest sugar crops in the world, amounting to 20% of the total worldwide sucrose production, second only to sugar cane. It is a significant source of sucrose in the temperate parts of the northern hemisphere [2,3]. Its byproducts can also serve as raw materials for producing other commodities, such as bioethanol and animal feed [4]. Due to their self-incompatibility, most commercial sugar beet varieties are hybrids descended from a single cytoplasmic male sterility (CMS) lineage, resulting in a more restricted genetic base [5–7]. However, introducing new characters into CMS seed parents is challenging [8]. Understanding the genetic diversity of sugar beet germplasm resources and population structure for sugar beet breeding is critical. The crops may lose some genetic diversity found in their wild ancestors due to the widespread adoption of genetically consistent variations [9–11]. The acquisition, evaluation, conservation, and use of crop germplasm resources are integral components of agricultural research and development since they serve as the basis for

theoretical biological investigations and molecular genetics as well as for the cultivation and choice of crop varieties.

The presence of diversity among various organisms, also known as biodiversity, can be observed at three distinct levels: genetic, species, and ecological. Genetic diversity is the term used to describe the overall genetic variation among individuals of different species within or between populations. This phenomenon is an inherent characteristic of living organisms that emerges from evolutionary processes [12]. Plant species require a wide range of genetic pools with high genetic variability to survive and adapt to the environment [13]. Sufficient genetic diversity is necessary to produce new species with desirable traits [14]. The probability of selecting high-quality germplasm from the germplasm resources available is enhanced as genetic diversity increases. To identify high-quality germplasms for breeding, studying genetic diversity can provide valuable insights into the evolution of genes. Crop genetic diversity can be established using lineage analysis, phenotypic evaluation, biochemical analysis, or molecular markers [15,16]. Phenotypic identification is the easiest and most fundamental research method for analyzing genetic diversity. Its application is intuitive and facilitates the quick understanding of crop genetic variation by analyzing phenotypic traits. However, certain drawbacks are associated with phenotypic identification, such as instability and vulnerability to environmental factors [17,18]. Molecular markers are extensively employed in population genetics due to their utility in evaluating genetic variation. They provide a direct measure of the extent of genetic variation between genotypes at the level of the DNA and establish a connection between phenotypic and genotypic variation [19–21]. Simple sequence repeats (SSRs) are repeating patterns ranging from 1 to 6 bp generally present in eukaryotic organisms' genomes [22] and are frequently utilized in plant breeding as marker types. SSR markers are commonly used to assess crop genetic diversity due to their consistency with Mendelian co-dominant inheritance, good stability, reproducibility, convenience, and widespread availability within the genome [10,23–26]. Various studies revealed that these markers have also been extensively utilized in the gene flow and genetic analysis of sugar beet populations [27,28]. Based on PCR amplification, insertion–deletion (InDel) markers are a type of base sequence length polymorphism markers. These are useful for determining the relationship between samples and allow the filtering out of single-base nonsense mutations. InDel molecular markers have proven effective in plant genetic diversity, population structure analysis, and assisted breeding. This is because they are highly reproducible, have co-occurring inheritance, and provide extensive genome coverage. These markers are also less expensive to develop and easier to design and detect [29–31].

Integrating phenotypic identification and molecular markers can enhance the accuracy of genetic diversity studies. This is because molecular markers solely identify variations at the DNA level, which may not necessarily manifest in observable traits (phenotypes) [32–34]. To select excellent hybrid combinations and obtain high-quality innovative varieties, it is necessary to identify and analyze sugar beet germplasm resources. This will further elucidate their germplasm's genetic background and intrinsic genetic structure. Using 27 and 33 pairs of SSR and InDel primers, respectively, in addition to 20 phenotypic traits such as leaf shape and color, the genetic diversity and population structure of 115 sugar beet polyembryonic pollinated lines and seven pairs of sugar beet monoembryonic sterile and maintained lines were examined in this study. This research, therefore, provides resources and theoretical direction for the future selection and breeding of sugar beet germplasm resources.

## 2. Materials and Methods

### 2.1. Plant Materials and Experimental Design

A total of 129 sugar beet lines, comprising seven pairs of sugar beet CMS and maintained lines as well as 115 polyembryonic pollinated lines, were utilized (Table S1). All the materials were backbone parents that can be used in breeding and were supplied by the Key Laboratory of Sugar Beet Genetic Breeding (Heilongjiang University, Harbin, China).

The test lines were grown at the experimental base of Heilongjiang University's Hulan Campus, where the soil was flat and black. This location is known for its mesothermal continental monsoon climate. Observation and data collection of agronomic qualities were conducted on ten plants from each line. The experiment was set up in randomized blocks, with two rows planted in each experimental region, a row length of 10 m, a spacing of 0.67 m, and a sowing depth of 3 cm. The aboveground section of the beet leaf cluster during its period of rapid growth was selected, specifically the largest true leaf, excluding the old and young leaves. The data of the traits in the belowground portion of the sugar beets were recorded throughout the sugar beets' sucrose accumulation stage, commonly referred to as the harvesting period. The test lines matched the criteria of the trait survey and exhibited satisfactory progress throughout the growth cycle.

*2.2. Phenotypic Trait Measurements*

The phenotypic traits of sugar beet germplasm were qualified using the "*Descriptors and date standard for beet*(*Beta vulgaris* L.)" reference [35]. The following parameters were examined in the 129 sugar beet lines: sixteen qualitative traits (leaf shape, color, margin shape, and surface, mesophyll thickness, leaf hairiness, petiole color and thickness, fascicled leaf type, root length, width, length-to-width ratio, shape, and groove depth, crown size, and skin roughness) and four quantitative traits (petiole width and length, plant height, and width of leaf coverage). Ten randomly selected plants were examined from each line for the above traits.

*2.3. DNA Extraction and Genotyping*

Total DNA was extracted from young sugar beet leaves at the 2–3 pairs of true leaf stage using the Cetyl Tri-methyl Ammonium Bromide (CTAB) method [36]. DNA purity and concentration were detected using a NanoDrop 2000/2000c Ultra-Micro UV-Vis Spectrophotometer (Thermo Fisher, Madison, WI, USA). The samples were diluted to 10 ng/µL to prepare the working solution and stored at −20 °C until further use.

A total of 129 sugar beet germplasms were examined as the test materials, from which 6 distinct germplasms were selected randomly as templates. These templates were then used to screen 42 SSRs and 41 pairs of InDel primers. After evaluating these primers for distinct bands, stable amplification, and significant polymorphisms, 27 pairs of SSR and 33 pairs of InDel primers were chosen for subsequent investigations (Table S2). Some of the SSR markers came from earlier research [25,37], while the remainder were designed and supplied by the Molecular Genetics Laboratory of Heilongjiang University using the sugar beet's whole genome sequence. All the InDel markers were designed and provided by the Molecular Genetics Laboratory (Heilongjiang University) using the sugar beet's whole genome sequence. Shanghai Bioengineering Co., Ltd. (Shanghai, China) developed the above-mentioned primers.

The 5 µL PCR amplification system contained 2.5 µL of 2 × Taq PCR Master Mix (BioTeke Corporation, Wuxi, China), 0.4 µL of primers, 1.1 µL of double distilled water (ddH$_2$O), and 1 µL of sugar beet genomic DNA. A Veriti 96-Well Thermal Cycler (ThermoFisher Scientific™, Shanghai, China) was used to perform PCR. The length, primer concentration, and base composition all affect the annealing temperature. To increase the PCR reaction's specificity and decrease non-specific binding, distinct PCR programs were employed for each primer, given that primers can only show amplified bands at their ideal annealing temperature. The steps in the PCR reaction were as follows: pre-denaturation at 94 °C for 3 min, followed by 35 cycles of 15 s at 94 °C, 15 s at 58 °C, 30 s at 72 °C, and a final extension at 72 °C for 5 min. The touchdown program was used for some primers: pre-denaturation at 94 °C for 3 min, followed by 15 s at 94 °C, annealing at 65 °C for 15 s, followed by two cycles of 65–56 °C for every one degree down to 56 °C, and extension at 72 °C for 30 s. After that, 20 cycles of 15 s at 94 °C, 15 s at 55 °C, 30 s at 72 °C, and a final extension at 72 °C for 5 min. The PCR products were then separated using 8% non-denaturing polyacrylamide gel electrophoresis, which was run for 1.5 h at a constant

180 V. The gel was stained with the non-toxic G-Red nucleic acid dye (BioTeke Corporation, Wuxi, China) and was photographed using a gel imager.

*2.4. Data Analysis*

A binary data matrix was created using the 0/1 assignment method. The amplified bands were manually read, and their types were recorded. A "1" was assigned to a band at a specific point, while a "0" was given to no band. The following genetic diversity indices were calculated using PopGene32 version 1.32 [38]: observed number of alleles (Na), effective number of alleles (Ne), observed heterozygosity (Ho), expected heterozygosity (He), Shannon's information index (I), Nei's expected heterozygosity, and gene flows (Nm). The gene diversity and polymorphism information content (PIC) [39] among different populations were calculated using PowerMarker 3.25 [40]. The STRUCTURE software was used to conduct the population structure analysis of the sugar beet germplasm resources [41], calculating the optimal number of subpopulations. This model-based software uses a Bayesian clustering method [42]. The Markov chain Monte Carlo (MCMC) technique computed the posterior probabilities. The MCMC chains were run using a model that allowed for admixture and correlated allele frequencies, with a 100,000 burn-in period followed by 100,000 iterations. Ten runs for each K-value were performed with K ranging from 1 to 10 to obtain an accurate estimation of the number of populations. The optimal number of subpopulations of the population was later determined by the rate of change in the a posteriori probability values ($\Delta K$) [42] using the web-based program STRUCTURE HARVESTER [43]. By combining the SSR and InDel data a clustered dendrogram of the 129 sugar beet genotypes was generated based on Nei's genetic distance [44], using the unweighted pair–group method with arithmetic averaging (UPGMA) as implemented using the MEGA7 [45].

All phenotypic trait data were assembled in Microsoft Excel. Basic statistical parameters were computed for each trait, including the minimum and maximum value, the mean, the standard deviation (SD), and the coefficient of variation (CV) [46]. The Shannon–Wiener diversity index (H′) [47,48] was calculated and correlated via the IBM Statistical Package for the Social Sciences (SPSS) Statistics, version 20.0. Origin 2020 was used for the principal component analysis (PCA) [49].

## 3. Results

This section presents the results of the genetic diversity analysis, population structure, genetic distance, cluster structure, and phenotypic trait variation in 129 sugar beet germplasms.

*3.1. Genetic Diversity Using SSR Primers*

With the use of 27 pairs of carefully selected SSR primers, a total of 129 sugar beet germplasm resources were thoroughly analyzed to identify any potential polymorphisms. A total of 130 alleles were found, ranging from 3 to 9 alleles per primer, with an average of 4.815 (Table 1). The mean Ne value was 3.054, with primer 26319 having the highest effective number of alleles at 4.870. The I ranged from 0.787 to 1.810, with a mean value of 1.231. The Ho of the primers was 1, and the minimum value was 0.781. The maximum value of He was 0.798, while the minimum was 0.524. Nei's expected heterozygosity ranged from 0.52 to 0.795. The average value of Nm was 0.599. The variation in the gene diversity index ranged from 0.529 to 0.798, with an average of 0.668. The PIC ranged from 0.419 to 0.773, with an average of 0.610. Primer 26319 demonstrated the maximum PIC value, while the minimum value was observed in primer 2170.

**Table 1.** Genetic diversity was amplified by 27 pairs of SSR primers in the 129 sugar beet germplasms.

| Locus | Na | Ne | I | Ho | He | Nei's Expected Heterozygosity | Nm | Genetic Diversity | PIC |
|---|---|---|---|---|---|---|---|---|---|
| 14118 | 4 | 3.116 | 1.224 | 1.000 | 0.682 | 0.679 | 0.660 | 0.684 | 0.624 |
| 17623 | 5 | 3.124 | 1.349 | 0.961 | 0.683 | 0.680 | 0.603 | 0.680 | 0.640 |
| 2305 | 5 | 2.941 | 1.237 | 0.946 | 0.663 | 0.660 | 0.632 | 0.660 | 0.602 |
| 11965 | 5 | 4.104 | 1.455 | 0.985 | 0.759 | 0.756 | 0.466 | 0.756 | 0.713 |
| 27374 | 4 | 2.560 | 1.080 | 0.891 | 0.612 | 0.609 | 0.637 | 0.615 | 0.544 |
| BQ588629 | 5 | 3.325 | 1.339 | 0.992 | 0.702 | 0.699 | 0.611 | 0.699 | 0.651 |
| L37 | 4 | 2.765 | 1.114 | 1.000 | 0.641 | 0.638 | 0.904 | 0.638 | 0.566 |
| L7 | 6 | 3.655 | 1.498 | 0.953 | 0.729 | 0.726 | 0.441 | 0.735 | 0.699 |
| 7492 | 4 | 2.595 | 1.097 | 0.929 | 0.617 | 0.615 | 0.634 | 0.632 | 0.574 |
| L47 | 3 | 2.702 | 1.042 | 0.992 | 0.632 | 0.630 | 0.927 | 0.630 | 0.555 |
| L57 | 6 | 2.492 | 1.107 | 0.977 | 0.601 | 0.599 | 1.107 | 0.599 | 0.523 |
| L35 | 5 | 3.196 | 1.264 | 0.935 | 0.690 | 0.687 | 0.413 | 0.713 | 0.663 |
| L48 | 4 | 2.878 | 1.182 | 0.953 | 0.655 | 0.653 | 0.639 | 0.658 | 0.600 |
| L64 | 4 | 3.247 | 1.240 | 0.898 | 0.695 | 0.692 | 0.444 | 0.697 | 0.638 |
| W21 | 5 | 2.372 | 1.078 | 0.781 | 0.581 | 0.578 | 0.380 | 0.615 | 0.561 |
| 2170 | 3 | 2.090 | 0.787 | 0.906 | 0.524 | 0.522 | 1.418 | 0.529 | 0.419 |
| 15915 | 5 | 2.773 | 1.182 | 0.960 | 0.642 | 0.639 | 0.561 | 0.665 | 0.610 |
| 17923 | 8 | 3.445 | 1.468 | 0.977 | 0.713 | 0.710 | 0.552 | 0.710 | 0.664 |
| 24552 | 5 | 2.825 | 1.128 | 0.961 | 0.649 | 0.646 | 0.682 | 0.652 | 0.582 |
| 26319 | 9 | 4.870 | 1.810 | 0.953 | 0.798 | 0.795 | 0.364 | 0.798 | 0.773 |
| 57236 | 4 | 3.315 | 1.249 | 0.954 | 0.701 | 0.698 | 0.538 | 0.698 | 0.637 |
| SSD6 | 6 | 3.302 | 1.450 | 0.992 | 0.700 | 0.697 | 0.617 | 0.697 | 0.661 |
| TC94 | 3 | 2.492 | 0.982 | 1.000 | 0.601 | 0.599 | 1.267 | 0.599 | 0.516 |
| BVV23 | 3 | 2.950 | 1.090 | 0.961 | 0.664 | 0.661 | 0.666 | 0.661 | 0.587 |
| W15 | 4 | 2.655 | 1.074 | 0.921 | 0.626 | 0.623 | 0.625 | 0.635 | 0.562 |
| BVV21 | 4 | 2.587 | 1.080 | 0.952 | 0.616 | 0.613 | 0.656 | 0.636 | 0.571 |
| TC122 | 7 | 4.079 | 1.627 | 0.985 | 0.758 | 0.755 | 0.469 | 0.755 | 0.722 |
| Average | 4.815 | 3.054 | 1.231 | 0.952 | 0.664 | 0.662 | 0.599 | 0.668 | 0.610 |

Na: Observed number of alleles; Ne: effective number of alleles; I: Shannon's information index; Ho: observed heterozygosity; He: expected heterozygosity; Nm: gene flow estimated from Fst = 0.25 (1 − Fst)/Fst; and PIC: polymorphism information content.

### 3.2. Genetic Diversity Using InDel Primers

Polymorphism was assessed on 129 sugar beet germplasm resources using 33 pairs of InDel primers. A total of 107 alleles were identified, with the number of detectable alleles varying from 2 to 5 per primer and an average of 3.242 (Table 2). The mean Ne value was 2.298, with ND129 having the highest value at 3.008. The I ranged from 0.268 to 1.175 (mean = 0.886). The maximum value of Ho of the primers was 0.985, while the minimum was 0.057. The maximum He value was 0.670, while the minimum was 0.140. Nei's expected heterozygosity ranged from 0.140 to 0.668. The mean value of gene flow was 0.660. The variation in the gene diversity index ranged from 0.140 to 0.668, with an average of 0.543. The PIC ranged from 0.130 to 0.602, with an average of 0.462. Primers ND129 and ND18 demonstrated the maximum and minimum PIC values.

**Table 2.** Genetic diversity was amplified by 33 pairs of InDel primers in the 129 sugar beet germplasms.

| Locus | Na | Ne | I | Ho | He | Nei's Expected Heterozygosity | Nm | Genetic Diversity | PIC |
|---|---|---|---|---|---|---|---|---|---|
| ND31 | 3 | 2.486 | 1.002 | 0.734 | 0.600 | 0.598 | 0.380 | 0.598 | 0.531 |
| ND75 | 3 | 1.876 | 0.729 | 0.636 | 0.469 | 0.467 | 0.533 | 0.467 | 0.378 |
| ND109 | 3 | 2.269 | 0.898 | 0.892 | 0.561 | 0.559 | 0.982 | 0.559 | 0.465 |

**Table 2.** *Cont.*

| Locus | Na | Ne | I | Ho | He | Nei's Expected Heterozygosity | Nm | Genetic Diversity | PIC |
|-------|------|-------|-------|-------|-------|------|-------|------|-------|
| ND113 | 3 | 1.825 | 0.663 | 0.682 | 0.454 | 0.452 | 0.768 | 0.452 | 0.353 |
| ND220 | 3 | 2.768 | 1.053 | 0.813 | 0.641 | 0.639 | 0.341 | 0.639 | 0.562 |
| ND223 | 3 | 1.750 | 0.702 | 0.558 | 0.430 | 0.429 | 0.467 | 0.429 | 0.360 |
| ND253 | 5 | 2.424 | 1.042 | 0.859 | 0.590 | 0.588 | 0.636 | 0.588 | 0.508 |
| ND258 | 3 | 2.856 | 1.074 | 0.954 | 0.652 | 0.650 | 0.688 | 0.650 | 0.576 |
| ND275 | 4 | 2.204 | 0.925 | 0.852 | 0.548 | 0.546 | 0.808 | 0.546 | 0.459 |
| ND277 | 3 | 1.804 | 0.791 | 0.566 | 0.447 | 0.446 | 0.435 | 0.446 | 0.404 |
| ND10 | 4 | 1.985 | 0.741 | 0.758 | 0.498 | 0.496 | 0.735 | 0.496 | 0.384 |
| ND11 | 3 | 1.808 | 0.658 | 0.659 | 0.449 | 0.447 | 0.701 | 0.447 | 0.351 |
| ND18 | 2 | 1.162 | 0.268 | 0.057 | 0.140 | 0.140 | 0.015 | 0.140 | 0.130 |
| ND108 | 4 | 2.844 | 1.115 | 0.914 | 0.651 | 0.648 | 0.566 | 0.648 | 0.576 |
| ND120 | 3 | 2.015 | 0.716 | 0.969 | 0.506 | 0.504 | 6.319 | 0.504 | 0.381 |
| ND121 | 3 | 2.579 | 1.004 | 0.876 | 0.615 | 0.612 | 0.629 | 0.612 | 0.530 |
| ND129 | 4 | 3.008 | 1.175 | 0.845 | 0.670 | 0.668 | 0.431 | 0.668 | 0.602 |
| ND142 | 3 | 2.428 | 0.956 | 0.954 | 0.590 | 0.588 | 1.070 | 0.588 | 0.499 |
| ND143 | 4 | 1.581 | 0.696 | 0.426 | 0.369 | 0.368 | 0.345 | 0.368 | 0.336 |
| ND162 | 4 | 2.577 | 1.052 | 0.984 | 0.614 | 0.612 | 0.942 | 0.612 | 0.533 |
| ND229 | 3 | 2.062 | 0.762 | 0.985 | 0.517 | 0.515 | 5.382 | 0.515 | 0.398 |
| ND231 | 4 | 2.931 | 1.104 | 0.930 | 0.661 | 0.659 | 0.600 | 0.659 | 0.586 |
| ND233 | 3 | 2.711 | 1.046 | 0.915 | 0.634 | 0.631 | 0.658 | 0.631 | 0.558 |
| ND243 | 5 | 2.701 | 1.104 | 0.915 | 0.632 | 0.630 | 0.663 | 0.630 | 0.564 |
| ND244 | 3 | 2.913 | 1.084 | 0.954 | 0.659 | 0.657 | 0.662 | 0.657 | 0.583 |
| ND246 | 2 | 1.928 | 0.674 | 0.667 | 0.483 | 0.481 | 0.564 | 0.481 | 0.365 |
| ND267 | 3 | 2.606 | 1.018 | 0.977 | 0.619 | 0.616 | 0.879 | 0.616 | 0.540 |
| ND270 | 4 | 2.325 | 0.959 | 0.868 | 0.572 | 0.570 | 0.799 | 0.570 | 0.479 |
| ND279 | 3 | 2.226 | 0.873 | 0.922 | 0.553 | 0.551 | 1.140 | 0.551 | 0.451 |
| ND280 | 2 | 1.555 | 0.542 | 0.434 | 0.358 | 0.357 | 0.388 | 0.357 | 0.293 |
| ND281 | 3 | 2.806 | 1.065 | 0.930 | 0.646 | 0.644 | 0.614 | 0.644 | 0.571 |
| ND283 | 2 | 1.961 | 0.683 | 0.813 | 0.492 | 0.490 | 1.062 | 0.490 | 0.370 |
| ND286 | 3 | 2.865 | 1.074 | 0.961 | 0.654 | 0.651 | 0.705 | 0.651 | 0.576 |
| Average | 3.242 | 2.298 | 0.886 | 0.796 | 0.545 | 0.543 | 0.660 | 0.543 | 0.462 |

Na: Observed number of alleles; Ne: effective number of alleles; I: Shannon's information index; Ho: observed heterozygosity; He: expected heterozygosity; Nm: gene flow estimated from Fst = 0.25 (1 − Fst)/Fst; and PIC: polymorphism information content.

### 3.3. Population Structure Analysis

A total of 129 sugar beet germplasms were examined for their population genetic structure using 27 and 33 pairs of SSR and InDel primers, respectively. The findings indicated that the maximum value of ΔK occurred at K = 3 (Figure 1), indicating the possibility of identifying three distinct populations from the 129 sugar beet germplasm resources (Figure 2). Fifty-one germplasms were grouped in Pop1, and all of them were polyembryonic pollinated lines. Fifty-seven germplasms were grouped in Pop2, and all of them were polyembryonic pollinated lines. Pop3 consisted of a total of twenty-one germplasms, which included both CMS lines and maintained lines.

### 3.4. Genetic Distance and Cluster Structure

The genetic distances (Nei's genetic distance) of the 129 germplasms, measured by 60 pairs of SSR and InDel molecular markers, varied from 0.099 to 0.466 (mean = 0.283) (Table S3). The findings indicated that 66 and 67 exhibited the lowest genetic distances, suggesting their closest proximity, while 29 and 129 had the highest genetic distances, marking the furthest separation. The 129 sugar beet germplasms could be distinguished from one another using the SSR and InDel markers, according to the findings of the UPGMA cluster analysis (Figure 3). These resources were grouped into three major taxa, which could be related to the findings of the population structure analysis.

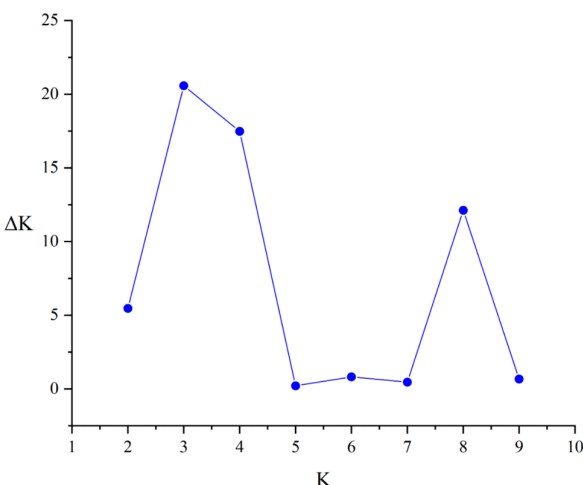

**Figure 1.** ΔK value change graph. The corresponding ΔK statistics identify the optimal STRUCTURE assignment of the 129 sugar beet germplasms (K = 3).

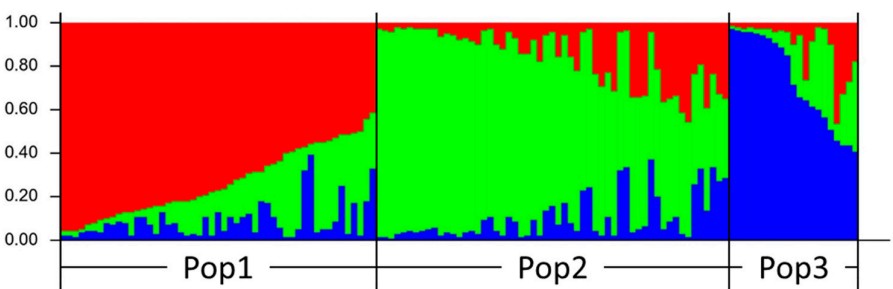

**Figure 2.** Population genetic structure of 129 sugar beet germplasms at K = 3. The highest Delta K is obtained at K = 3, suggesting that the population could be divided into three subpopulations: Pop1, Pop2, and Pop3.

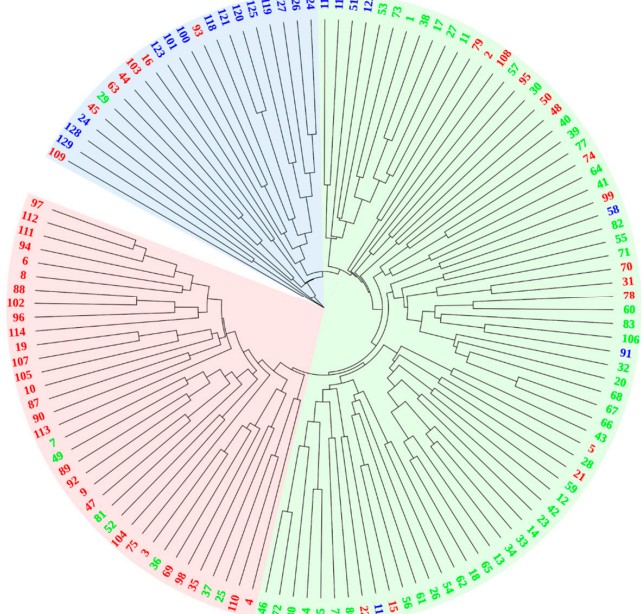

**Figure 3.** The 129 sugar beet germplasms clustered using UPGMA, based on SSR and InDel markers. The materials are categorized into three taxa (indicated by blue, green, and red colors). The color of the number labels corresponds to the taxon findings of the population structure analysis.

### 3.5. Variation Analysis of the Phenotypic Traits

For each of the four qualitative traits, a statistical analysis of the 129 germplasms' phenotypic characteristics was carried out (Table 3). The results demonstrated that petiole length (17.11%) and width (16.64%) exhibited the highest coefficients of variation, while plant height (9.73%) was the lowest. Out of the four characteristics evaluated, only the plant height coefficient of variation had a value below 10%. The findings indicated a relatively low level of variability in plant height among the test lines. Conversely, the remaining three qualitative characteristics showed significant variations across the genetic components.

**Table 3.** Statistical analysis of variation in measured traits of sugar beet germplasm resources.

| Trait | Max (cm) * | Min (cm) ** | Average (cm) | SD | CV (%) |
|---|---|---|---|---|---|
| Petiole width | 2.760 | 1.075 | 1.596 | 0.266 | 16.64 |
| Petiole length | 38.940 | 18.030 | 26.663 | 4.562 | 17.11 |
| Plant height | 69.450 | 41.625 | 53.566 | 5.211 | 9.73 |
| Width of leaf coverage | 103.260 | 54.620 | 79.442 | 11.990 | 15.09 |

* Maximum values; ** minimum values; SD: standard deviation; and CV: coefficient of variation.

The lines exhibited variable degrees of different descriptive traits, as determined by the frequency analysis of 16 descriptive traits in the 129 germplasms (Table 4). Petiole color had the highest number of variation types among the six primary variation forms. The petiole color percentages were highest for the "white-green" variety at 60.47% and the "light green" variety at 27.13%. Root length was the second most common attribute, with five distinct types of variation. Medium (37.21%) and long root (31.01%) lengths exhibited the maximum percentages. The lines had many of the same leaf properties, such as being primarily green in color, having large waves at the edges of the leaf margins, and having slightly wrinkled leaf surfaces. Less hairy (96.90%) leaves were more common than hairier (3.10%) ones, whereas a thin (58.91%) mesophyll was more prevalent than a medium (33.33%) or thick (31.01%) mesophyll. The petiole thickness was most commonly medium, and 61.24% of the fascicled leaves were erect. Most of the roots were medium in length, broad in width, and high in their length-to-width ratio. A total of 51.94% of the roots were cuneiform; the crown size was primarily small; the root groove depth was generally not apparent; and the epidermis was very smooth.

**Table 4.** Frequency distribution and diversity index of descriptive characteristics of sugar beet germplasm resources.

| Traits | Characteristic Description (Proportion of Distribution, %) | | | | | | H′ |
|---|---|---|---|---|---|---|---|
| | 1 | 2 | 3 | 4 | 5 | 6 | |
| Leaf shape | Halberd 23.26 | Share 38.76 | Tongue 37.98 | | | | 1.074 |
| Leaf color | Light green 20.93 | Green 43.41 | Dark green 35.66 | | | | 1.057 |
| Leaf margin shape | Full margin 5.43 | Small wave 29.46 | Medium wave 26.36 | Big wave 38.76 | | | 1.237 |
| Leaf surface | Smooth 2.33 | Wavy 23.26 | Slight crease 44.96 | More creases 29.46 | | | 1.146 |
| Mesophyll thickness | Thin 58.91 | Medium 33.33 | Thick 31.01 | | | | 1.041 |
| Leaf hairiness | Little 96.90 | Much 3.10 | | | | | 0.138 |
| Petiole color | White 3.88 | White-green 60.47 | Light green 27.13 | Green 6.98 | Pink 0.00 | Purplish red 1.55 | 1.034 |
| Petiole thickness | Slight 28.68 | Medium 45.74 | Thick 25.58 | | | | 1.065 |

**Table 4.** *Cont.*

| Traits | Characteristic Description (Proportion of Distribution, %) | | | | | | H' |
|---|---|---|---|---|---|---|---|
| | 1 | 2 | 3 | 4 | 5 | 6 | |
| Fascicled leaf type | Erect 61.24 | Semi-crawl 34.11 | Crawl 4.65 | | | | 0.810 |
| Root length | Very short 3.10 | Short 13.18 | Medium 37.21 | Long 31.01 | Very long 15.50 | | 1.395 |
| Root width | Narrow 26.36 | Medium 32.56 | Broad 41.09 | | | | 1.082 |
| Root length-to-width ratio | Small 30.23 | Medium 22.48 | Large 47.29 | | | | 1.051 |
| Root shape | Cuneiform 51.94 | Conical 31.78 | Spindle 16.28 | | | | 1.000 |
| Crown size | Small 43.41 | Medium 33.33 | Large 23.26 | | | | 1.068 |
| Root groove depth | None 27.91 | Not obvious 37.98 | Shallow 28.68 | Deep 5.43 | | | 1.240 |
| Skin roughness | Very smooth 49.61 | Smoother 38.76 | Very rough 11.63 | | | | 0.965 |

H': Shannon–Wiener diversity index.

Among the 16 descriptive traits, the genetic diversity indices were ranked as follows: root length (1.395) > root groove depth (1.240) > leaf margin shape (1.237) > leaf surface (1.146) > root width (1.082) > leaf shape (1.074) > crown size (1.068) > petiole thickness (1.065) > leaf color (1.057) > root length-to-width ratio (1.051) > mesophyll thickness (1.041) > petiole color (1.034) > root shape (1.000) > skin roughness (0.965) > fascicled leaf type (0.810) > leaf hairiness (0.138).

*3.6. Correlation Analysis*

The correlation study of 129 sugar beet germplasm resources revealed seven pairs of highly significant correlations between traits. Five of these pairs were positively associated, and two were negatively correlated. The analysis was based on four qualitative and sixteen descriptive traits (Table 5, Table S4, and Figure 4).

The results of the correlation analysis of the four quantitative traits showed that petiole width was highly significant and positively correlated with petiole length (0.361 **), plant height (0.392 **), and leaf cover width (0.441 **). The petiole length was highly significant and positively correlated with plant height (0.788 **) and leaf cover width (0.642 **), and plant height was highly significant and positively correlated with leaf cover width (0.444 **).

The leaf margin shape showed a significant negative association with the root length-to-width ratio and a highly significant negative relationship with the petiole thickness among the 16 qualitative features. A significant positive correlation was detected between leaf blade thinness, petiole thinness, and root epidermal texture. Moreover, a highly significant positive correlation was also detected with the root furrow depth, and a significant negative correlation was identified with the leaf clump shape. The petiole thickness and the root shape showed a significant negative association; however, the root epidermal texture showed a significant positive correlation. A significant negative correlation was observed between the petiole thickness and the leaf margin shape. A significant positive correlation was identified between the root length-to-width ratio and the root length (correlation coefficient = 0.661), and a highly significant negative correlation was observed with the root width. The root shape and width demonstrated a significant positive correlation (correlation coefficient = 0.267). Root groove depth was significantly negatively correlated with mesophyll thickness, crown size, and skin roughness, with correlation coefficients of 0.322, 0.273, and 0.386, respectively.

**Table 5.** Correlation analysis of four quantitative characteristics of sugar beet germplasm resources.

| Trait | Petiole Width | Petiole Length | Plant Height |
|---|---|---|---|
| Petiole length | 0.361 ** | | |
| Plant height | 0.392 ** | 0.788 ** | |
| Width of leaf coverage | 0.441 ** | 0.642 ** | 0.444 ** |

** $p < 0.01$.

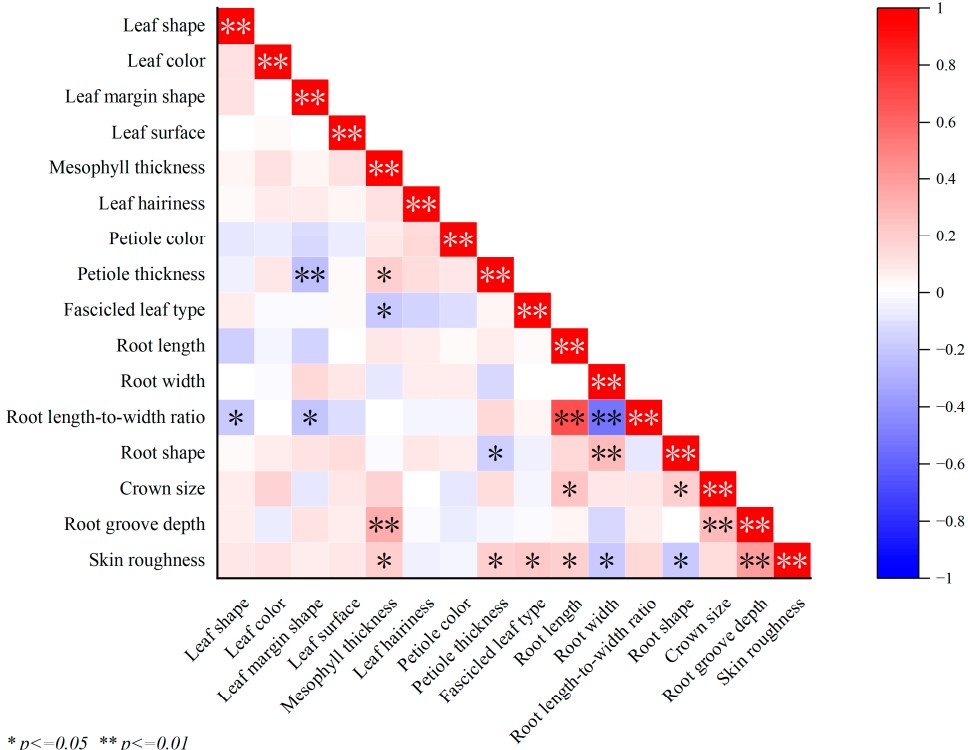

* $p<=0.05$  ** $p<=0.01$

**Figure 4.** Heatmap of quality trait correlations.

### 3.7. PCA Evaluation

PCA was performed on 20 traits of 129 sugar beet germplasms (Table 6, Figures 5 and 6). The first five major components of sugar beet, with a cumulative contribution rate of 51.151%, encompass the primary variable information of the correlated indices. The initial eigenvalue of principal component 1 (PC1) was 3.615, with a contribution rate of 18.075%. The main agronomic traits in PC1 were the petiole length, the width of leaf coverage, and the plant height, with characteristic vectors of 0.434, 0.422, and 0.373, respectively. These were classified as the overall aboveground plant proportion. The initial eigenvalue of principal component 2 (PC2) was 1.903, with a contribution rate of 9.516%. The primary agronomic traits in PC2 were the roots' length (0.431), the crown's size (0.367), and the petiole's thickness (0.330). The initial eigenvalue of principal component 3 (PC3) was 1.853, with a contribution rate of 9.263%. PC3's principal agronomic characteristics had characteristic vectors of 0.354, 0.333, and 0.329 for the root groove depth, the mesophyll thickness, and the leaf shape, respectively. The initial eigenvalue of principal component 4 (PC4) was 1.480, with a contribution rate of 7.399%. The root length, the root shape, and the fascicled leaf type were the primary agronomic features in PC4, with corresponding characteristic vectors of 0.391, 0.389, and 0.277, respectively. The initial eigenvalue of principal component 5 (PC5) was 1.380, with a contribution rate of 6.899%. The fascicled leaf type (0.511), the skin roughness (0.429), and the plant height (0.258) were the primary agronomic traits.

**Table 6.** Eigenvectors and cumulative contribution rates of each principal component.

| Principal Component Number | Eigenvalue | Contribution Rate (%) | Cumulative Contribution Rate (%) |
|---|---|---|---|
| 1 | 3.615 | 18.075 | 18.075 |
| 2 | 1.903 | 9.516 | 27.590 |
| 3 | 1.853 | 9.263 | 36.854 |
| 4 | 1.480 | 7.399 | 44.252 |
| 5 | 1.380 | 6.899 | 51.151 |
| 6 | 1.173 | 5.866 | 57.017 |
| 7 | 1.072 | 5.359 | 62.376 |
| 8 | 1.068 | 5.340 | 67.716 |
| 9 | 0.968 | 4.838 | 72.555 |
| 10 | 0.850 | 4.252 | 76.807 |
| 11 | 0.804 | 4.020 | 80.827 |
| 12 | 0.691 | 3.454 | 84.281 |
| 13 | 0.678 | 3.389 | 87.669 |
| 14 | 0.620 | 3.102 | 90.771 |
| 15 | 0.466 | 2.328 | 93.099 |
| 16 | 0.434 | 2.172 | 95.271 |
| 17 | 0.395 | 1.977 | 97.249 |
| 18 | 0.295 | 1.476 | 98.725 |
| 19 | 0.142 | 0.710 | 99.435 |
| 20 | 0.113 | 0.565 | 100.000 |

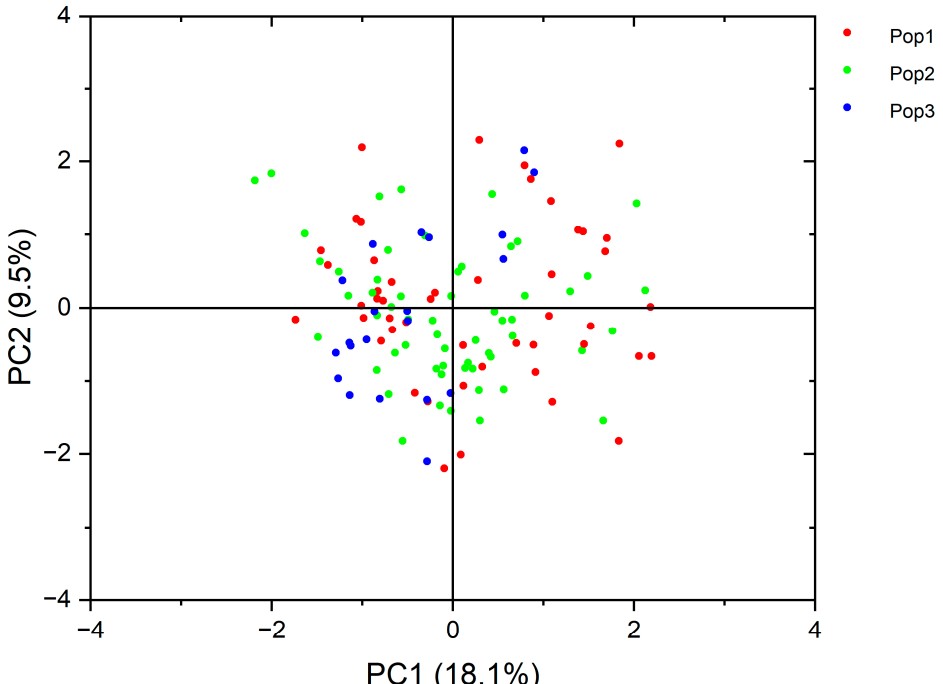

**Figure 5.** Principal component analysis (PCA) of 129 sugar beet germplasms based on 20 phenotypic traits. The coloration of each germplasm is determined by the population structure results (Pop1 = red, Pop2 = green, and Pop3 = blue). Populations one and two show significant overlap, indicating a relatively high genetic similarity.

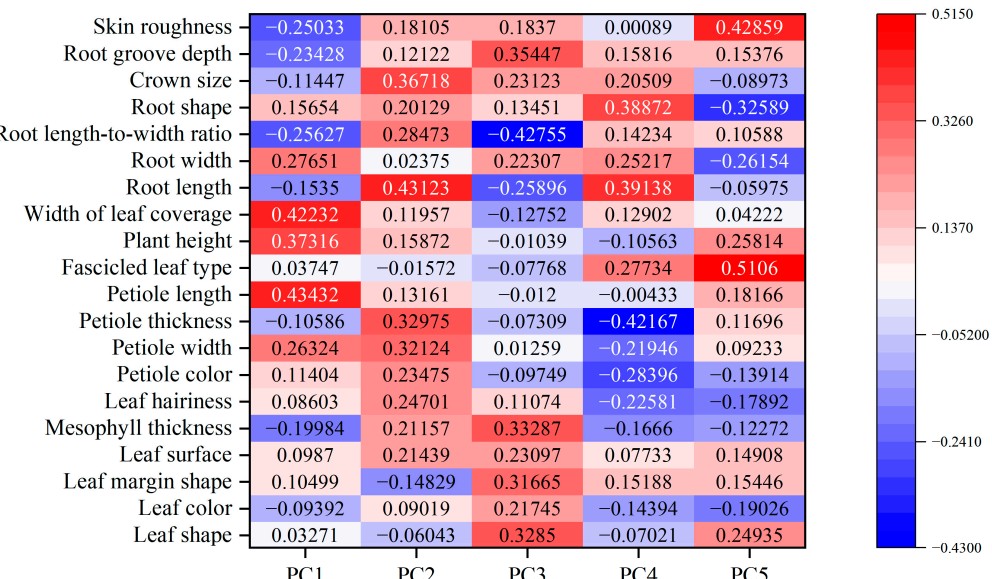

**Figure 6.** PCA heatmap of sugar beet germplasms.

## 4. Discussion

The development and utility of superior sugar beet germplasm resources are critical for effective breeding. The germplasms of sugar beets are essential for scientific study and breeding and for developing novel, high-yield, high-quality, and disease-resistant cultivars. Genetic variety typically ensures biodiversity. However, only a high diversity of genes can guarantee biodiversity's continuous survival. The collection, breeding, conservation, genetic diversity analysis, and study of sugar beet germplasm resources are of great value in all developed nations. Crop genetic diversity is best investigated via joint usage of morphological and molecular markers. Kleine et al. previously used morphological and molecular methods to examine the potential of 30 different nematode-resistant lines as a new source of resistance [50]. In another investigation, eight pairs of amplified fragment length polymorphism (AFLP) markers and four root traits were used to examine the molecular and morphophysiological properties of wild and cultivated sugar beet lines from the Italian Adriatic coast [51]. Taguchi et al. investigated the genetic diversity of 63 of the best Japanese sugar beet inbred lines using 33 cleaved amplified polymorphic sequences (CAPS). In their study, the contribution of genetic and molecular characteristics to the phenotypic variance of agronomically significant traits was evaluated using 38 SSR markers [52]. The genetic diversity of sugar beet in Kazakhstan was determined using randomly amplified polymorphic DNA (RAPD) markers, geomorphological traits, root yield, and sugar content [53].

The molecular markers we selected were not specific enough to identify specific traits and only reflected variation at the DNA level that was not necessarily expressed in the phenotype, so the relationship between traits and markers was not further analyzed. The use of a combination of phenotypic identification and molecular markers has been widely employed to analyze genetic diversity. Morphological markers were the first genetic markers to be used for genetic diversity analyses and have the advantage of being simple and intuitive, but are susceptible to environmental influences and less stable. In contrast, the use of molecular markers better reflects the genetic differences of germplasm resources and makes up for the shortcomings of morphological identification, and the combination of the two methods can improve the resolution of genetic diversity analysis. This whole study is assisting breeding work as, before configuring hybrid combinations, it is necessary to understand the phenotypic trait characteristics of existing sugar beet polyembryonic germplasms to screen sugar beet germplasm with excellent target traits and provide a basis

for the innovation of sugar beet germplasm resources and varietal selection and breeding, avoiding the blindness of traditional formulation methods of hybrid combinations.

SSR markers have a more comprehensive range of applications and are more common and polymorphic. The amplification product band patterns of InDel markers are clearer and more accurate than those of SSR markers, preventing errors brought on by specificity and complexity. Given the advantages of these two markers, we decided to combine them to improve the efficacy of this study and yield superior and reliable findings when examining the genetic diversity of sugar beet germplasm. As PIC may precisely reflect the marker polymorphisms in the population under investigation, it is an essential tool in genetic research. The investigation results showed that PIC values higher than 0.50 were present in 68.33% of the 60 highly polymorphic molecular marker pairs. Collectively, the tested primers were polymorphic and extremely valuable. At the molecular level, the 129 sugar beet germplasm samples showed a significant degree of genetic variation. However, the PIC values of the SSR markers were greater than those of the InDel markers, indicating that the microsatellites were genotypically diverse and highly informative. STRUCTURE was used to divide the population into subgroups with distinct structures. The results demonstrated that the 129 germplasms were categorized into three groups. The germplasms were divided into three groups using cluster analysis, and all CMS and maintained lines were grouped into a single category. This was in line with the findings of the population structure research. The average genetic distance between the sterile lines and the maintained and polyembryonic sugar beet germplasm was 0.315 when the genetic distance results were combined. These findings suggest that the genetic correlation between the seven pairs of sterile lines and the maintained and polyembryonic sugar beet germplasm resources was relatively distant. Therefore, in future breeding efforts, superior germplasms can be screened in each population based on the findings of the sugar beet germplasm population structure, and, by combining the findings of genetic distance analysis, hybrid combinations can be configured to predict hybrid advantage.

The genetic variation indices of 16 qualitative qualities varied from 0.138 to 1.395 regarding phenotype. Almost every trait in the thirteen traits was covered by more than one genetic diversity index. Leaf hairiness was the only trait with a lower genetic diversity index. A variation coefficient of more than 10% was shown by three of the four quantitative parameters, indicating that the sugar beet germplasm resources had a high breeding potential and were genetically rich. The four qualitative traits demonstrated a highly significant positive correlation, and the descriptive traits also showed varying degrees of correlation. These results indicated mutual promotion and constraints among the different traits of sugar beet lines. The indirect selection of target traits could potentially be based on a correlation analysis. The correlation between traits should be fully considered during sugar beet cultivation to promote the coordinated development of each trait [17]. The PCA results demonstrated that the first five principal components reflected the most significant information in the 20 phenotypic traits with a cumulative contribution of 51.151%. The section of the plant aboveground was designated as the first principal component. The size and shape of the roots and leaves were represented in the second and third principal components. The overall shape of the plant's aboveground and belowground portions was primarily expressed in the fourth and fifth principal components.

This study provided a theoretical foundation for germplasm innovation and variety selection. Further selection of genetically distant germplasms for hybridization can improve sugar beet genetic diversity by introducing excellent sugar beet germplasm resources worldwide. This strategy will enhance germplasm resources with increased genetic information for the better development and utilization of sugar beet lines.

## 5. Conclusions

This study examined 60 pairs of molecular markers and 20 phenotypic traits to determine the genetic diversity of sugar beet germplasms in China. The findings demonstrated that sugar beet germplasms displayed high genetic variation at the molecular level. More-

over, due to the superior resolution of the SSR markers over the InDel markers, they were deemed more appropriate for genetic diversity research. Rich levels of genetic variation in phenotypic traits were also seen in the sugar beet germplasm, suggesting a rich genetic background in Chinese sugar beet germplasm.

The characteristics of sugar beet germplasm can be considered through breeding identification and genetic diversity analyses of sugar beet germplasm resources. This will help filter out the best sugar beet germplasm, lay the groundwork for selecting superior hybrid combinations and obtaining high-quality innovative varieties, and provide theoretical references for future breeding work. This is crucial in accelerating the production of high-yield high-quality sugar beet in China and achieving sustainable development in the sugar beet industry.

**Supplementary Materials:** The following supporting information can be downloaded at https://www.mdpi.com/article/10.3390/horticulturae10020120/s1: Table S1: Sugar beet (*Beta vulgaris* L.) germplasm resources used in the study; Table S2: The DNA molecular markers used in the study; Table S3: Genetic Distance of 129 sugar beet germplasms; Table S4: Correlation analysis of 16 descriptive characteristics of sugar beet germplasms.

**Author Contributions:** Conceptualization, F.P. and Z.W.; methodology, Z.W.; software, F.P.; validation, F.P., S.L., Z.P. and Z.W.; formal analysis, F.P.; investigation, F.P.; resources, S.L., Z.P. and Z.W.; data curation, F.P. and Z.W.; writing—original draft preparation, F.P.; writing—review and editing, S.L., Z.P. and Z.W.; visualization, F.P.; supervision, Z.W.; project administration, Z.W.; funding acquisition, S.L. and Z.W. All authors have read and agreed to the published version of the manuscript.

**Funding:** This research was funded by the Special Fund for the Improvement of High-quality Sugar Beet Varieties of the National Sugar Modern Agricultural Industrial Technology System grant number CARS-170111 and the Basic Scientific Research Projects of provincial colleges and universities in the Heilongjiang Province grant number 2022-KYYWF-1039.

**Data Availability Statement:** The original contributions presented in the study are included in the article/Supplementary Materials, further inquiries can be directed to the corresponding author/s.

**Conflicts of Interest:** The authors declare no conflicts of interest.

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
