# Peer review of "Genetic Diversity and Population Structure Analysis of Excellent Sugar Beet (Beta vulgaris L.) Germplasm Resources"

_horticulturae, doi:10.3390/horticulturae10020120_

Round 1

Reviewer 1 Report

Comments and Suggestions for Authors

The manuscript titled “Genetic diversity and population structure analysis of excellent sugar beet (Beta vulgaris L.) germplasm resources" aims to investigate the genetic diversity and population structure of sugar beet germplasm (115 polyembryonic pollinated lines and seven pairs of sugar beet monoembryonic sterile and maintained lines) using 27 and 33 pairs of SSR and InDel primers, respectively, in addition to 20 phenotypic traits such as leaf shape and color. While the topic could be relevant and of general interest to the journal's readership, several concerns must be addressed before publication.

·         The authors are strongly advised to carefully review the manuscript to address grammar and other editing issues.

·         Use keywords that are different from the title words.

·         In the Material and Methods section, it is important to include or complete the sources of all chemicals, software, and equipment by adding the city, state, and country information. This additional detail provides readers with specific information about where these items were sourced, ensuring transparency and facilitating reproducibility.

·         At the initial reference, linking any abbreviation with its complete name is imperative for the reader to easily comprehend your communication, such as CMS in line 85, ALFP in line 272, and CAPS in line 274, even if you think they are general terms, not all the readers familiar with them.

·         Line 111, “ng/L” should be “ng/µl”

·         Within the Material and Methods section, the authors must either furnish a suitable citation or offer a more comprehensive description of the methodology. This will ensure the possibility of replicating the procedure.

·         In Tables 1, 2, 3, and 4, explain all abbreviated terms in the respective table footnotes, ensuring readers can comprehend and easily follow the content.

·         Figure 1 and 2: Enhance the figure caption by incorporating additional details to render it more self-explanatory.

·         Figure 2, add what the X and Y axes are.

·         In the caption of Figure three, what is Figure 129?

·         I recommend you present the PCA results as a graph, not a Table.

Comments on the Quality of English Language

Moderate editing of English language required.

Reviewer 2 Report

Comments and Suggestions for Authors

The study entitled:” Genetic diversity and population structure analysis of excellent sugar beet (Beta vulgaris L.) germplasm resources”, submitted by Peng et al. fits perfectly within the scope of the journal “Horticulturae”. This study investigated the genetic diversity, population structure, and cluster analysis of 1,129 sugar beet germplasm resources using molecular markers. The primary aim was to assess the genetic diversity and population structure for identifying superior germplasm suitable for breeding. The findings offer a theoretical foundation for the future selection and breeding of superior sugar beet germplasm resources. The paper is well-written, and well-structured. However, some minor remarks must be taken into account before acceptance of this work.

Abstract:

·         Line 1: Instead of "1 129," it should be "1,129" for correct formatting.

·         Line 11: Consider rephrasing to avoid repetition. For example, "The most appropriate K value, indicating three populations (K = 3), was revealed by the population structure analysis results."

·         Line 15: What do the authors mean by "efficients” ? I think it should be "coefficients".

Introduction and Materials and Methods sections

·         Line 26: Specify "world sucrose production" as either "worldwide sucrose production" or "global sucrose production" for clarity.

·         Line 30: Instead of "hybrids that descended from a single CMS," consider "hybrids descended from a single CMS lineage."

·         Line 83: Consider specifying the source or origin of the 129 sugar beet lines for transparency. You can point that this information is given Table S1 (in the supplementary file).

·         Line 122: Clarify the rationale for using different PCR annealing temperatures for different primers.

·         Line 141: add more details about (STRUCTURE) used for population structure analysis. Same goes for all the softwares used in this study (STRUCTURE HARVESTER; MEGA 7 …)

·         Line 144: What do you mean by "relatedness" in the context of creating a clustered dendrogram.

Results and Discussion section

·         Line 154: Provide a brief overview or introduction to the Results section. For example, "In this section, we present the results of genetic diversity analysis, population structure, genetic distance, cluster structure, and phenotypic trait variation in 129 sugar beet germplasms."

·         Consider grouping the statistics into sub-sections for better organization. For example, "3.1 Genetic Diversity Using SSR Primers" and "3.2 Genetic Diversity Using InDel Primers."

·         Lines 180-187: When mentioning genetic distances, it might be helpful to specify the unit of measurement (e.g., Nei's genetic distance). Also, consider using subheadings for SSR and InDel markers.

·         The correlation analysis section is informative, but certain sentences lack clarity. For example, in Line 224, "correlation with one another".

Conclusion and future perspectives

·         I suggest rephrasing the conclusion as it should succinctly summarize key findings and their implications for future research or applications.

·         Avoid reiteration of information already presented; instead, focus on synthesizing the main outcomes.

General comments:

·         Generally, SSR markers exhibit widespread distribution throughout the entire genome, displaying high polymorphism and reliability. Numerous studies have extensively documented genetic analyses, such as population structure and phylogeny, using SSR markers. While the rationale behind the selection of SSR markers is well-founded, the decision to incorporate InDel polymorphic molecular markers in the study is worth further explaining in the manuscript. The authors' choice to include InDel markers in their investigation prompts an inquiry into the specific advantages or unique insights that InDel markers might offer in comparison to SSR markers.

·         What insights can be gained from analyzing the population structure of sugar beet germplasm, and how does it influence breeding strategies?

·         Are there distinct subgroups within the sugar beet germplasm, and how do these subgroups relate to geographical or ecological factors?

Reviewer 3 Report

Comments and Suggestions for Authors

This is a very interesting study using molecular markers.

1. The distinction between qualitative and quantitative traits does not seem clear.

2. Please clearly describe the development method or developer for the marker used.

3. It does not appear that the relationship between traits and markers has been studied. What is the meaning of analyzing diversity indices differently depending on the trait?

Round 2

Reviewer 1 Report

Comments and Suggestions for Authors

Thank you. The Author responded to all of my comments!

Comments on the Quality of English Language

Minor editing of English language required.

Author Response

Dear Reviewer:

Thank you for your comments concerning our manuscript entitled “Genetic diversity and population structure analysis of excellent sugar beet (Beta vulgaris L.) germplasm resources”. We have asked native speaker to edit the English language again. We really hope that the flow and language level have been improved.

Sincerely yours,

Fei Peng

Reviewer 3 Report

Comments and Suggestions for Authors

I recommend that the contents of "response 3" be added to the discussion section of the manuscript.

Author Response

Dear Reviewer:

Thank you for your comments concerning our manuscript entitled “Genetic diversity and population structure analysis of excellent sugar beet (Beta vulgaris L.) germplasm resources”. In response to your suggestion, we have added the contents of "Response 3" to the discussion section of the manuscript. See them in lines 360-375.

Sincerely yours,

Fei Peng